# Integrated microcavity electric field sensors using Pound-Drever-Hall detection

Xinyu Ma[1], Zhaoyu Cai[2], Chijie Zhuang [1] ✉, Xiangdong Liu[1], Zhecheng Zhang[1], Kewei Liu[2], Bo Cao [2], Jinliang He [1], Changxi Yang [2], Chengying Bao [2] ✉ & Rong Zeng[1] ✉

Discerning weak electric fields has important implications for cosmology, quantum technology, and identifying power system failures. Photonic integration of electric field sensors is highly desired for practical considerations and offers opportunities to improve performance by enhancing microwave and lightwave interactions. Here, we demonstrate a high-Q microcavity electric field sensor (MEFS) by leveraging the silicon chip-based thin film lithium niobate photonic integrated circuits. Using the Pound-Drever-Hall detection scheme, our MEFS achieves a detection sensitivity of 5.2 $\mu$V/(m$\sqrt{\text{Hz}}$), which surpasses previous lithium niobate electro-optical electric field sensors by nearly two orders of magnitude, and is comparable to atom-based quantum sensing approaches. Furthermore, our MEFS has a bandwidth that can be up to three orders of magnitude broader than quantum sensing approaches and measures fast electric field amplitude and phase variations in real-time. The ultra-sensitive MEFSs represent a significant step towards building electric field sensing networks and broaden the application spectrum of integrated microcavities.

Electric field sensing is an essential technique in basic science with applications ranging from detecting cosmic fast radio burst[1] to understanding quantum physics[2] and lightning origin[3]. It is also at the heart of assuring smooth running of many aspects of the modern society[4]. For example, it is important to monitor the reliability of power systems[5], test the electromagnetic compatibility (EMC) in semiconductor foundries[6] and enable radar imaging for vehicles (see Fig. 1a). Many physical effects including electro-optical[7,8], piezoelectric[9], electrostatic force[10], and quantum effects[11,12] have been harnessed for electric field sensing. In particular, quantum sensing using Rydberg atoms[11] or trapped ions[12] has reached an unprecedented detection limit down to 5.5 $\mu$V/(m$\sqrt{\text{Hz}}$) and 0.2 $\mu$V/(m$\sqrt{\text{Hz}}$) in recent years, respectively. However, these approaches need a complicated system for quantum states preparation and the instantaneous sensing bandwidth is relatively narrow.

Electro-optical (Pockels) approaches are relatively simple and can reach high sensitivity and large bandwidth simultaneously. Optical interferometers made of electro-optical active materials have been widely used for electric field sensing, with lithium niobate (LN) being a material of keen interest[8]. This is due to its large Pockels coefficient, long term reliability, and low loss. LN sensors are photonic integration compatible and on-chip waveguides have been used to form these interferometers for electric field sensing recently[13]. Long waveguides and electrodes are needed to improve the sensitivity; thus, velocity mismatch between electric and optical fields will ultimately limit the sensitivity scaling. The associated large form factor may also limit the sensor spatial resolution. Furthermore, interferometer-based sensors have to operate around the quadrature point and ambient fluctuations impose practical issues on them[14].

Instead of long waveguides, light circulation in micro optical cavities may also be used to enhance the sensitivity with a compact

---

[1]State Key Laboratory of Power Systems, Department of Electrical Engineering, Tsinghua University, Beijing 100084, China. [2]State Key Laboratory of Precision Measurement Technology and Instruments, Department of Precision Instruments, Tsinghua University, Beijing 100084, China. ✉e-mail: chijie@tsinghua.edu.cn; cbao@tsinghua.edu.cn; zengrong@tsinghua.edu.cn

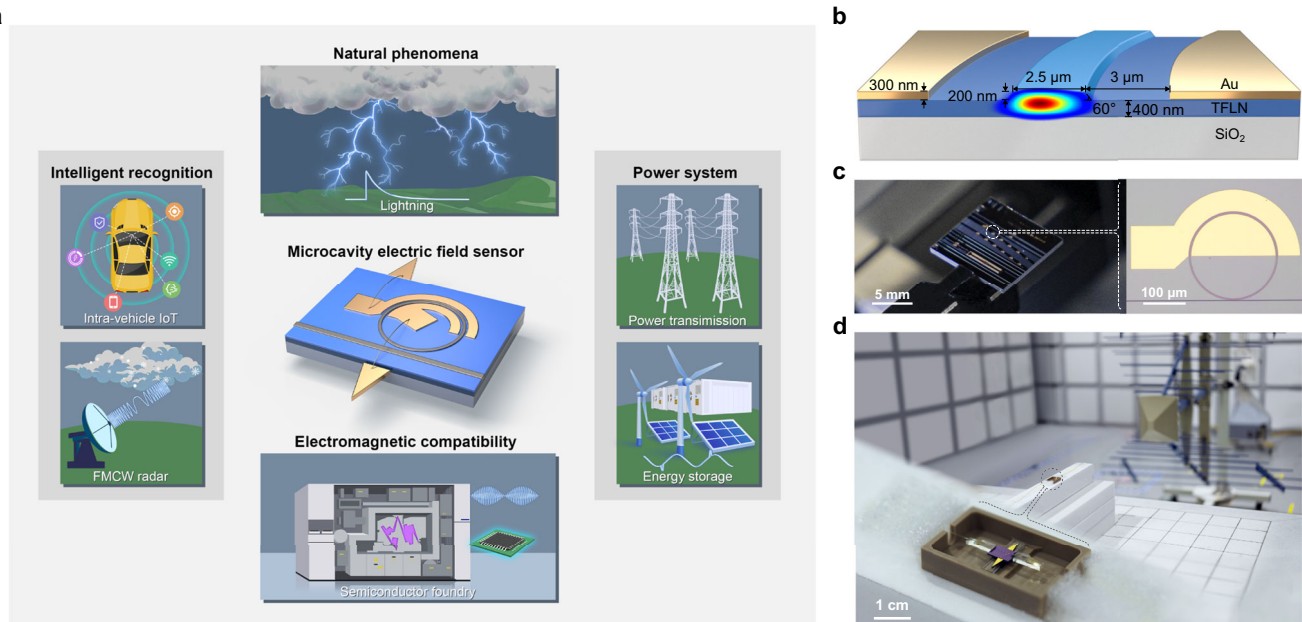

**Fig. 1 | Microcavity electric field sensor (MEFS) and potential applications.**
**a** MEFSs can be used in many scenarios ranging from radar, monitoring natural phenomena, power systems, to semiconductor foundry. **b** An illustration of the thin-film lithium niobate (TFLN) microresonator and the optical mode profile. **c** A photograph of the TFLN chip and a photograph of the microcavity under a microscope. **d** A picture of a packaged MEFS and our test environment in an anechoic chamber. Credit: image of the lithography system in panel **a** is adapted from ASML.

form factor. High-Q microcavities have been the drive for chip-based high-coherence lasers[15], classical and quantum microcombs[16,17]. And they have been widely used for ultrasensitive sensing of nanoparticles[18,19], acceleration[20], force[21] and rotation[22,23], etc. Another important application of high-Q optical cavities is to act as references to narrow laser linewidth down to sub-10 mHz, laying the foundation for the next generation timekeeping systems[24]. Such a highly efficient laser frequency purification relies upon the Pound-Drever-Hall (PDH) method[25] that converts frequency detuning from the cavity resonance into a steep error signal. The steep, linear error signal can also be used for sensing. Indeed, it has enabled ultrasensitive strain sensing using the cavity-like transmission spectrum of a fiber Bragg grating (FBG)[26].

Here, we combine the PDH scheme with high-Q thin-film lithium niobate (TFLN) microcavities[27,28] to demonstrate microcavity electric field sensors (MEFSs). TFLN has been an emerging platform for nonlinear photonics and power efficient electro-optical interface[28,29]. Bulk LN and TFLN (or heterogeneously integrated TFLN) microcavities have been used for electric field sensing by measuring optical power variation due to electric field induced resonance shift, reaching a detection sensitivity down to 0.9 mV/(m$\sqrt{\text{Hz}}$)[30–32]. The PDH scheme and the enhanced electro-optical interaction in our high-Q TFLN microcavity allow the integrated MEFSs to reach a sensitivity of 5.2 $\mu$V/(m$\sqrt{\text{Hz}}$), a sensing dynamic range of 123 dB, and a 3 dB bandwidth of 0.4 GHz. It also enables real-time measurement of various electric waveforms including double exponential waves and amplitude/phase modulated waves. A theoretical formula bearing the MEFS sensitivity and bandwidth is derived to guide further optimization. The formula can also be used for analysis of PDH detection-based sensors for other physical quantities. Our work sets a record sensitivity and dynamic range for LN-based electric field sensors and holds prospects for mass production.

## Results

### PDH sensing principle for MEFSs
Illustration of the MEFS is plotted in Fig. 1. TFLN on insulator was etched to create microring resonators (see Methods). Gold electrodes were deposited and patterned around the microcavity with a narrow gap of 3 $\mu$m, which enables efficient electro-optical interaction (Figs. 1b, c). Figure 1b also shows the cross-section geometry of the microcavity. The electrodes were bonded to a pair of dipole antenna[33], which has a length of 5 mm and a bottom width of 2 mm and may improve the detection sensitivity by about 24 dB (see Fig. 1a, d and Supplementary Sec. 3). Then the device was packaged by a polyetheretherketone box, which has a low dielectric constant and a low hygroscopicity, with a dimension of 4.5 × 2.9 × 0.8 cm. Four devices were packaged in total and the fiber-to-fiber insertion loss after packaging was about 10 dB (5 dB per fiber-to-chip facet).

The devices were tested in an experimental setup illustrated in Fig. 2a (see Methods). Two of the devices were measured to have intrinsic Q-factors of 2.0 million and 0.4 million after packaging as shown in Figs. 2b, c (denoted as Device 1 and Device 2). Device 1 and Device 2 have diameters of 400 $\mu$m and 200 $\mu$m, respectively. The width for the coupling waveguide is 1.2 $\mu$m, and the gap between the coupling waveguide and the microcavity is 1 (0.85) $\mu$m for Device 1 (2). The difference in the Q-factor is primarily attributed to the fact that they were fabricated and packaged in different batches, rather than the difference in the design.

To have the PDH signal, we phase modulated the input laser (an external cavity diode laser, ECDL) at a frequency of $\Omega_m/2\pi$ around 3 GHz, and the measured PDH error signal when scanning the laser across the resonance is shown in Fig. 2d. A linear error signal with a slope of 2.24 V/GHz was observed for Device 1, while this slope was 0.55 V/GHz for Device 2, due to the relatively low Q-factor. When locking the laser to the resonance with a low bandwidth, the laser cannot track the fast resonance variation due to the sensing electric field at a high frequency of $\Omega_s/2\pi$ and the resulting laser-resonance detuning creates a sensing signal (Fig. 2d). By contrast, the laser tracks the FBG reflection wavelength change induced by the slow strain signal (i.e., sensing frequency stays within the locking bandwidth) in ref. 26. In that scheme, the sensing signal is readout by beating the PDH locked

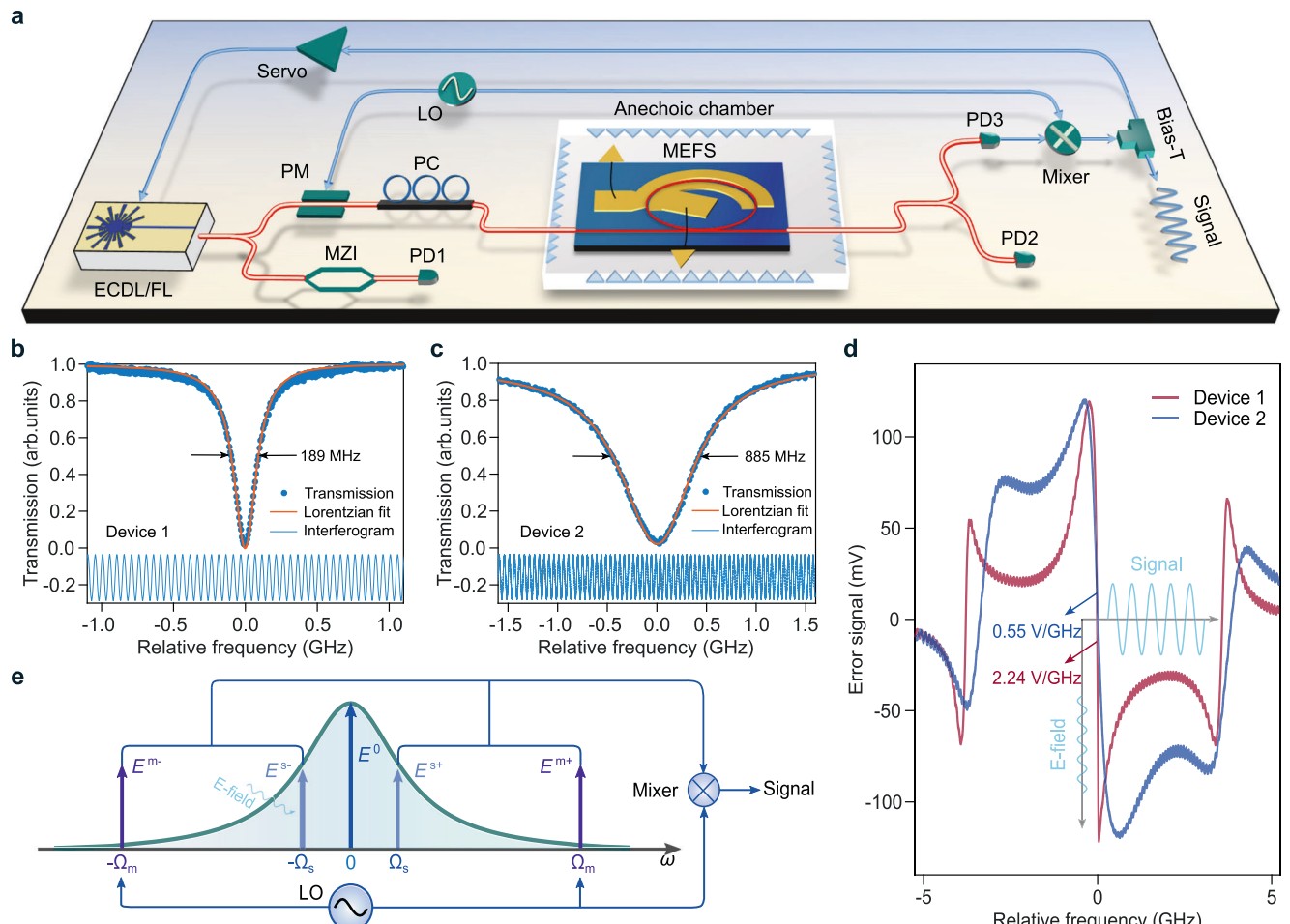

**Fig. 2 | Principle of microcavity electric field sensor (MEFS). a** Experiment setup for the MEFS characterization. ECDL: external cavity diode laser, FL: fiber laser, LO: local oscillator, PM: phase modulator, PC: polarization controller, MZI: Mach-Zehnder interferometer, PD: photodetctor. **b** Transmission of Device 1 showing an intrinsic (loaded) Q-factor of $2.0 \times 10^6$ ($1.0 \times 10^6$). **c** Transmission of Device 2 showing an intrinsic (loaded) Q-factor of $0.4 \times 10^6$ ($0.2 \times 10^6$). The interferograms in (**b**), (**c**) are the output from the MZI in (**a**). The interferograms have a free-spectral-range of 50 MHz, and are used to calibrate the laser frequency scan. **d** PDH

error signals for Device 1 and Device 2, which have a slope of 2.24 V/GHz and 0.55 V/GHz, respectively. The steep error signal can transform the electric field induced microcavity resonance shift into a sensing signal. **e** Modulation of the intracavity carrier field $E^0$ from the sensing electrical field generates sidebands $E^{s\pm}$ that are enhanced by the cavity resonance. The sensing signal results from beating between $E^{s\pm}$ and external phase modulation sidebands $E^{m\pm}$, after mixing with the local oscillator at $\Omega_m$. Beatnote between $E^{s\pm}$ and $E^{m\mp}$ should also generate sensing signal, but is omitted for clarity.

laser with a stabilized optical frequency comb (i.e., using frequency signal for sensing). Here, we use the error signal (voltage signal) as the output directly. The operation point for our MEFS can be precisely determined by locking to a zero error signal, whereas microcavity sensors relying upon direct power detection and interferometer-based sensors need careful calibration of a non-zero operation point. When using feedback loop to control the operation point, absolute optical power fluctuation will affect the operation point for those sensors but will not affect our PDH MEFS. The influence of the power fluctuation on the error signal slope can be normalized by a reference detector for our MEFS (e.g., PD2 in Fig. 2a).

In the frequency domain, the sensing principle can be understood as follows (Fig. 2e). External cavity phase modulation of the laser generate sidebands $E^{m\pm}$, which do not couple strongly to the cavity as $\Omega_m/2\pi$ is larger than the resonance linewidth. The carrier is coupled into the cavity ($E^0$), and is modulated by the microcavity electrode and the electric field at $\Omega_s$. The modulation leads to sidebands $E^{s\pm}$ that are enhanced by the high-Q microcavity. Beating between $E^{s\pm}$ (after coupling into the waveguide) and $E^{m\pm}$ will generate the sensing signal after mixing with the local

oscillator. By analyzing these sidebands, we derived the sensing signal for a relatively weak electric field as (see Fig. S1 and Supplementary Sec. 1 for the derivation),

$$V_{RF} = KP_{in}\frac{\kappa_e}{\kappa}\frac{\varepsilon}{\sqrt{(\kappa/2)^2 + \Omega_s^2}}\cos\left(\Omega_s t - \text{atan}\left(\frac{2\Omega_s}{\kappa}\right)\right), \quad (1)$$

where $\kappa = \kappa_0 + \kappa_e$ is the total loss rate of the cavity including the intrinsic loss rate $\kappa_0$ and the external coupling rate $\kappa_e$, $P_{in}$ is the off-resonance input power for the photodetector (PD), $K$ is a conversion coefficient including response from the external-cavity phase modulation, photodetector, mixer and electro-optical modulation efficiency of the microcavity electrode and the antenna, $\varepsilon$ is the sensing field. Therefore, high-Q TFLN MEFSs enjoy at least twofold benefits of efficient electro-optical modulation (large $K$) and strong cavity enhancement (small $\kappa$). Eq. (1) also shows high Q-factor improves the sensitivity, but at an expense of a reduced sensing bandwidth, as $E^{s\pm}$ needs to be hosted by the resonance. Furthermore, it is found that critical coupling ($\kappa_e = \kappa_0$) is the optimal coupling condition for higher MEFS sensitivity for $\Omega_s \ll \kappa$. And our microcavities have such a coupling condition (Figs. 2b, c).

## Sensing characterization of MEFSs

The MEFSs were tested in an anechoic chamber so as to shield against environmental electric fields (see Fig. 1d). The sensing electric field was generated from a log-periodic antenna or a horn antenna to access different sensing frequency $\Omega_s$, and the MEFSs were placed 1-3 m away from the antenna. The electric field was calibrated using a receiving log-periodic antenna and a probe kit (AR Inc., FL7006) with various experiment configurations. The emitting antenna was horizontally polarized, matching the polarization direction of the MEFS antenna. The sensing signal was recorded by an oscilloscope (recording $V_{RF}$) and an electrical spectrum analyzer (recording $\langle |V_{RF}|^2 \rangle$). An example of the sensor output is shown in Fig. 3a. Both devices recover a 100 MHz electric field with a high fidelity. Narrow radiofrequency (RF) tones with a high signal-to-noise ratio (SNR) can be observed in the electric spectrum with a 200 Hz resolution bandwidth (RBW) (Fig. 3b). The strength of the RF tones decreases with decreasing electric field. Such a decrease exhibits an excellent linearity (measured with a 1 Hz RBW, see Fig. 3c). The dynamic range of the linear regime is up to 123 dB and 122 dB for Device 1 and Device 2, respectively, which is the largest for electro-optical LN electric field sensors to our knowledge, see Supplementary Sec. 4. The observed saturation of the sensing signal and the upper limit of the linear range can be attributed to the non-negligible high-order intracavity sidebands and pump depletion at strong applied electric field (see Supplementary Sec. 1 and Fig. S2). When taking the intersection between the linear fit and the noise floor

as the detection sensitivity, our MEFS yields a sensitivity of 8.8 $\mu$V/(m$\sqrt{\text{Hz}}$) (29.5 $\mu$V/(m$\sqrt{\text{Hz}}$)) for Device 1 (2). The detection sensitivity is also reflected by the intercept of the log-log plot, which increases with the Q-factor quadratically after normalizing $P_{in}$, as predicted by Eq. (1) (see inset of Fig. 3c).

As an advantage of the PDH sensing scheme, cavity transmission of the carrier is suppressed for critically coupled microcavities and the noise floor is less impacted by the laser intensity noise (see Supplementary Sec. 2). The noise floor of the current MEFSs is strongly limited by the PD noise. Nevertheless, this floor around the testing 100 MHz rises due to the mixing between the relaxation oscillation of the ECDL around 2.9 GHz and $\Omega_m/2\pi$ (see Supplementary Sec. 2 and Fig. S3). By replacing the ECDL with a fiber laser, the noise floor around 100 MHz decreases by about 6 dB and a sensitivity of 5.2 $\mu$V/(m$\sqrt{\text{Hz}}$) can be attained. Since the fiber laser has a relatively narrow frequency tuning range, the used mode in this case has a slightly lower Q-factor than the ECDL case and the measured squares are slightly weaker than the circles in Fig. 3c. The sensitivity is a record for LN-based electric field sensors and is on par with the Rydberg atom result[11]. Further noise floor reduction may also contribute to a larger dynamic range.

The measured MEFS bandwidth also conforms to Eq. (1). We varied the frequency of the emitting antenna to measure the frequency response of our MEFS. The measured response can be fitted by a Lorentzian function, and the fluctuations may result from the non-uniform response of the emitting antenna in our anechoic chamber as

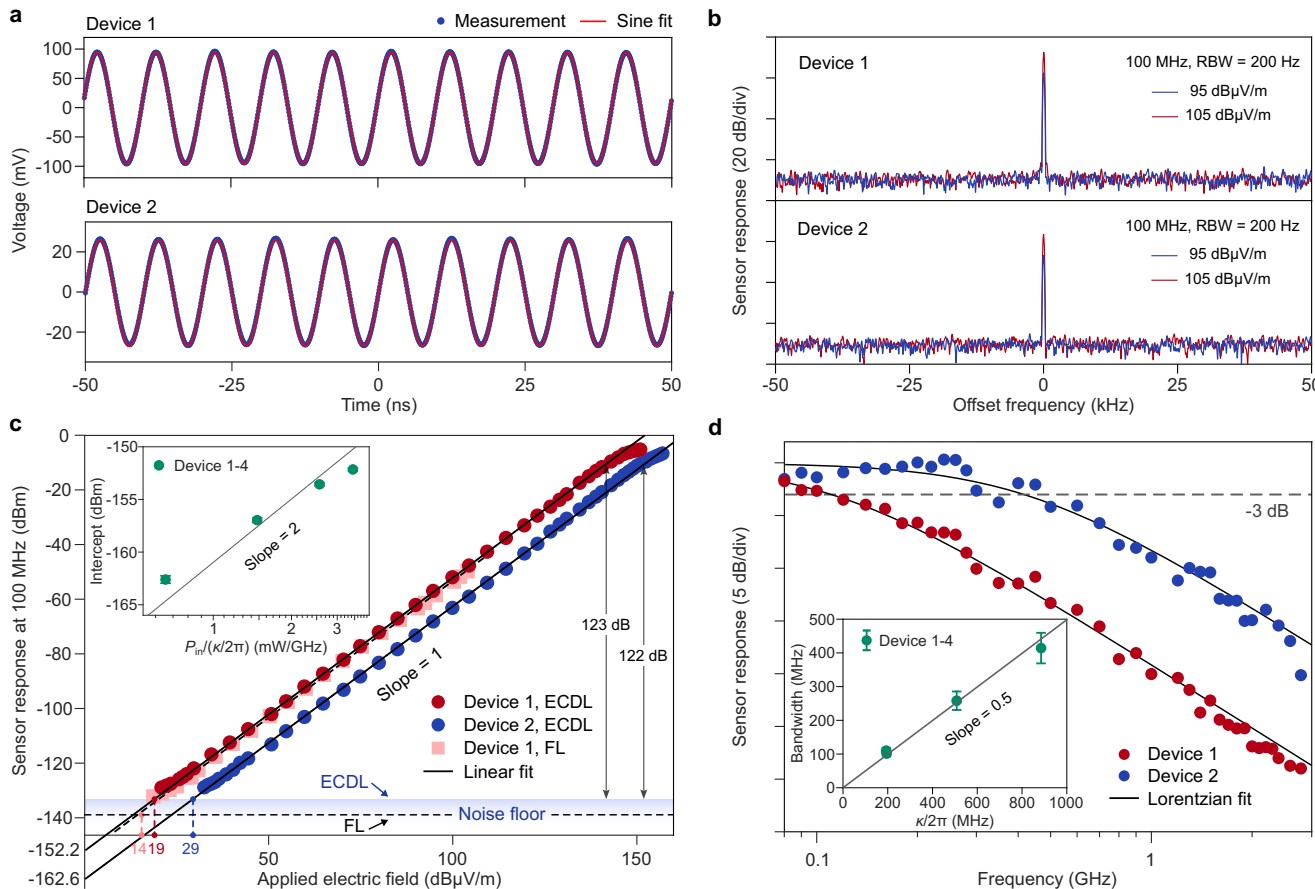

**Fig. 3 | Microcavity electric field sensor (MEFS) characterization. a** Temporal signal from the MEFSs (Device 1 and Device 2) when using a 100 MHz signal as the input. **b** Sensing signal in the electric spectrum with a resolution bandwidth (RBW) of 200 Hz. **c** Linearity measurement of the MEFSs. The measurement the sensitivity of the MEFS can reach 8.8 $\mu$V/(m$\sqrt{\text{Hz}}$) (29.5 $\mu$V/(m$\sqrt{\text{Hz}}$)) for Device 1 (2) when using an external cavity diode laser (ECDL). By replacing the ECDL with a fiber laser (FL), the noise floor can decrease and the sensitivity can reach 5.2 $\mu$V/(m$\sqrt{\text{Hz}}$). Device 1

(2) shows a linear dynamic range of 123 dB (122 dB). The inset shows the intercept of the log-log plot increases quadratically with the Q-factor. **d** The frequency response of our MEFSs exhibits a Lorentzian roll-off. The measured 3 dB bandwidth is 110 MHz (410 MHz) for Device 1 (2). The bandwidth is about a half of microcavity mode linewidth as shown in the inset. The error bar in the inset corresponds to the 95% confidence bounds of the Lorentzian fit.

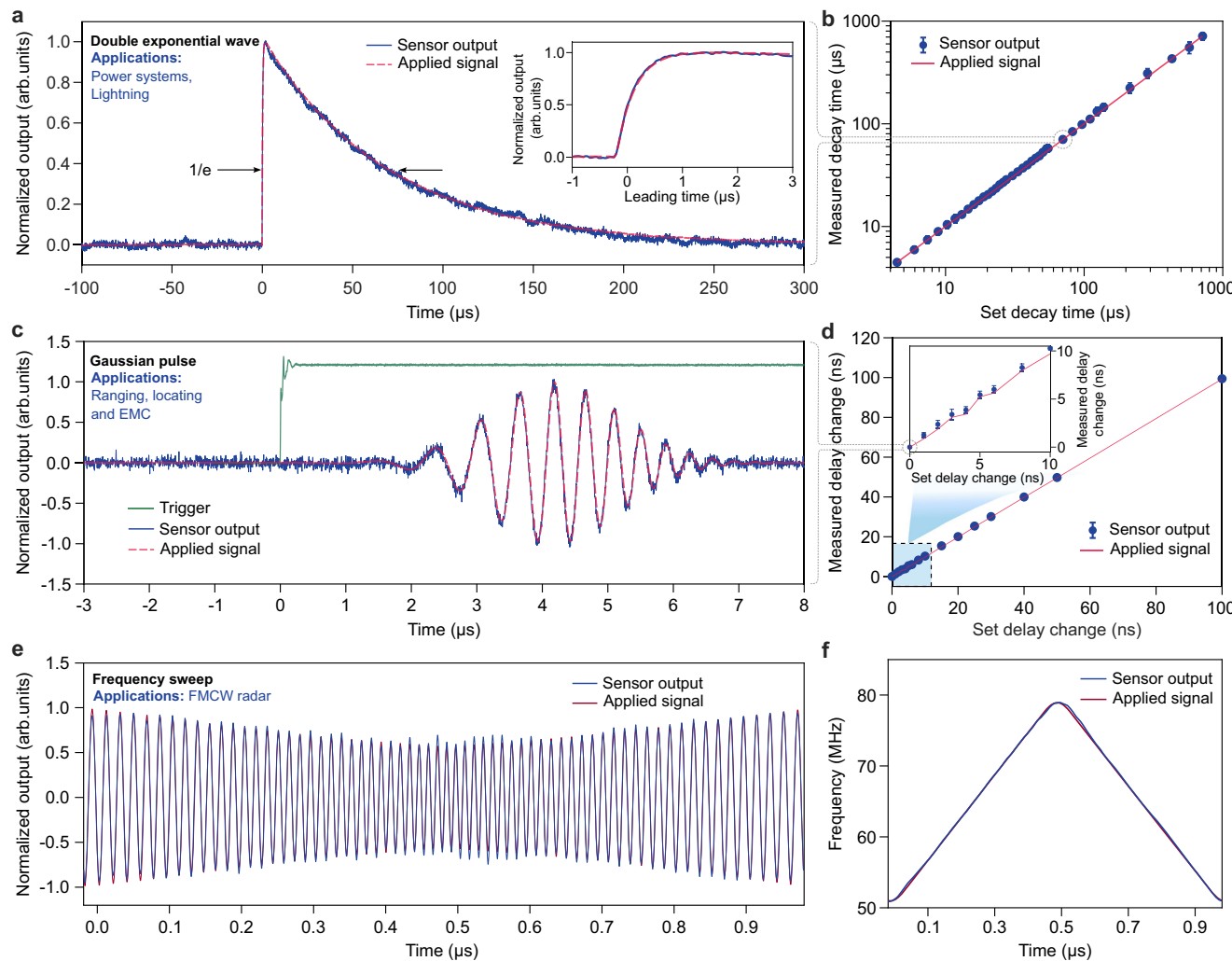

**Fig. 4 | Real-time measurement of RF waveforms. a** Measurement of a double exponential wave that represents a failure indicator for power systems and lightning induced electric field. The inset shows the measured sharp leading edge of the waveform. **b** Measured decay time agrees well with the set decay time. **c** Measurement of a Gaussian pulse by the MEFS and the green curve is a trigger signal from the function generator. **d**, The delay between the Gaussian pulse envelope peak and the trigger can be accurately measured by the MEFS. **e** Measurement of a frequency modulated continuous wave (FMCW) signal. **f** The retrieved instantaneous frequency follows the FMCW input signal closely. The error bar in (**b**), (**d**) corresponds to the standard deviation in multiple measurements.

well as the RF devices (Fig. 3d). The fitted 3 dB bandwidth is close to $\kappa/4\pi$ for all the 4 measured devices (see inset of Fig. 3d; two devices have similar Q-factors and overlap in the inset). Since the MEFS response is symmetric with respect to the zero frequency, the measured $\kappa/4\pi$ bandwidth is also consistent with Eq. (1). By interacting with another mode, electric field at higher frequencies can be measured. As an aside, the frequency response of the dipole antenna does not impact the MEFS bandwidth significantly (see Supplementary Sec. 3 and Fig. S4).

### Real-time measurements of electric fields
The relatively large sensing bandwidth (0.4 GHz for Device 2) enables us to measure different types of RF waveforms in real-time. Here the MEFS was tested in a parallel plate electric field generator. We show the measurement of a double exponential wave in Fig. 4a. This waveform corresponds to a failure indicator of power systems[5] and lightning electric field in nature[34]. Our MEFS captures the sharp leading edge (inset of Fig. 4a) and the slow decaying trailing tail successfully. When varying the 1/e decay time of the trailing tail from 4 $\mu$s to 500 $\mu$s, the fitted 1/e decay time is in excellent agreement with the input signal (Fig. 4b). The MEFS can also be used to measure pulses with a Gaussian

envelope (Fig. 4c). When using a trigger signal (green curve in Fig. 4c) for the function generator (Keysight 33612A) and the oscilloscope and changing the delay between the trigger the synthesized pulse, the fitted envelope peak location also agrees with the set delay (Fig. 4d). The capability to detect the RF pulse timing accurately may be used for ranging and locating electric grid fault[35]. The carrier frequency of the Gaussian pulse was actually time varying in the measurement, and our sensing signal reproduced it accurately.

This ability can also enable instantaneous phase and frequency measurement of the electric field; thus, MEFSs may be used for FMCW radars for distance and velocity measurements. To show this capability, we replaced the input by a periodic frequency sweeping signal (sweeping from 50 MHz to 80 MHz in 1 $\mu$s). The sensing signal is plotted in Fig. 4e and follows the input signal. The retrieved instantaneous frequency via the Hilbert transform is also in a close agreement with the set frequency sweeping pattern (Fig. 4f). The measured frequency range and chirping rate are limited by our function generator. In principle, the range can be comparable or slightly larger than the 3 dB bandwidth of the MEFS. Thus, Device 2 can allow tens cm-level resolution when used for FMCW radar.

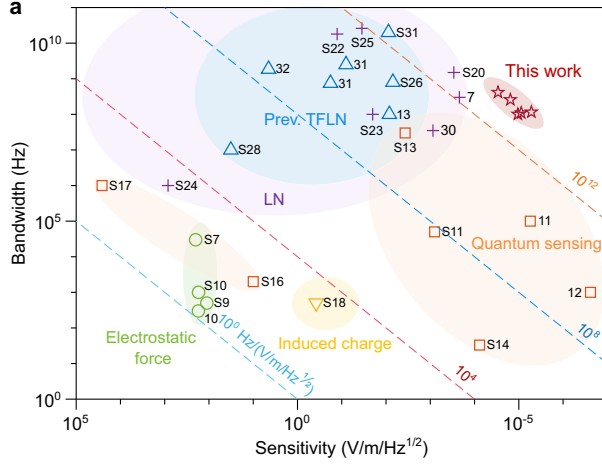

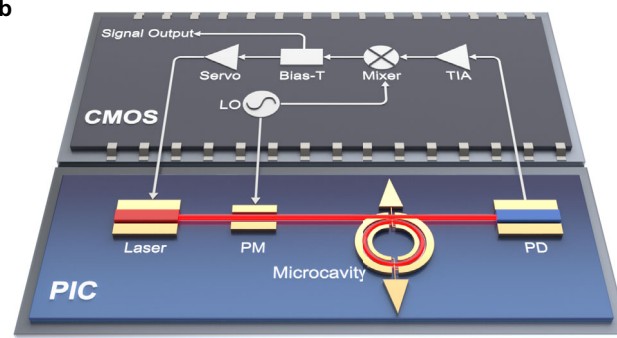

**Fig. 5 | Performance and integration conceptualization of a fully integrated microcavity electric field sensor (MEFS). a** Comparison of our MEFSs with other electric field sensors on sensitivity and bandwidth. The prefix S represents references in the Supplementary Information. **b** MEFSs can be integrated together with on-chip laser and photodetector in the future. PIC photonic, integrated circuits, TIA Transimpedance amplifier, PM phase modulator, PD photodetector, LO local oscillator, CMOS complementary metal-oxide-semiconductor.

## Discussion

Our MEFSs have a combination of high sensitivity and large bandwidth. We compare the performance with reported electric field sensors in Fig. 5a (see also Supplementary Sec. 4 and Table S1). Although the sensitivity is slightly lower than the trapped ions approach[12], our MEFSs have a much larger instantaneous bandwidth. The sensitivity may be further improved by increasing Q-factor. The Q-factor of integrated TFLN microcavities can reach 10 million[27]. Further optimization of the fabrication process can improve the Q-factor, and the absorption limited Q-factor for the TFLN platform is over 100 million[36]. Increasing $K$ can also improve the sensitivity without sacrificing the bandwidth. For example, the current electrodes only surround half the microcavity and adding electrodes for the remaining half should increase the response to $2K$. Racetrack microcavities with straight sections could further improve $K$ by always aligning the light polarization with the crystal orientation that gives the largest electro-optical response, while the relative orientation changes within a round trip for a ring microcavity. Theoretical analysis predicts that this alignment can give another twofold improvement[37], resulting in a conversion coefficient of $4K$. Since the noise floor is currently limited by the PD, reducing fiber-to-fiber insertion loss (for example, using the bilayer mode size converter design[38]) and increasing $P_{in}$ can also enhance the sensitivity (see Supplementary Sec. 2). Replacing the PD by one with a lower noise-equivalent-power (e.g., avalanche photodetector, APD) can also contribute to increasing the sensitivity. However, if laser noise dominates over PD noise, PD will no longer influence the

sensitivity. The noise floor will finally be limited by the shot noise of the detected light. The improved $K$ due to the narrow gap between the microcavity and the electrode should also be weighed against its collateral impact on Q-factor for optimal $K/\kappa$. Taking these together, the sensitivity of MEFSs may be pushed to a level better than the reported trapped ion result[12] with a much simpler architecture.

Our work further shows high-Q microcavities are a powerful engine for integrated photonics. The current progress on photonic integrated circuits bodes well for the full integration of the MEFS[28]. Heterogeneous integration of TFLN phase modulator and high-Q microcavity with III-V laser and PD[39,40] should be possible in the near future. The feasibility of compact CMOS circuit for PDH locking and readout has also been established[41]. Thus, chip-integrated MEFSs with an exceptional sensitivity, a broad bandwidth and a large dynamic range can be envisioned (see Fig. 5b). This integration can further improve the robustness, and reduce the size, weight and power consumption, making MEFSs better fit the cluttered environments. Ultimately, these scalable MEFSs can be deployed for laboratory and field measurements, for instance forming a sensing array in space to detect fast radio bursts[1] or to study laser-guided lightning behaviors[42]. Accurately detecting and locating lightning can also contribute to thunderstorm forecasting[43]. Integrated MEFSs may also enable comprehensive studies of high-altitude transient electromagnetic events in the stratosphere and mesosphere to reveal their impacts on lightning origin and ionosphere[44]. Additionally, they can serve as portable devices to ensure human health in specific work environments under electromagnetic exposure[45].

## Methods
### Fabrication and package of MEFS
The chip was fabricated on a 4-inch wafer with a 600 nm thick x-cut TFLN and 4.7 $\mu m$ thick buried-oxide layer on top of a silicon substrate (NanoLN). TFLN wafers with other thickness can also be used for MEFSs. The designed microcavities were patterned by DUV lithography. The pattern was transferred to TFLN layer using argon plasma in an inductively coupled plasma etching process, leaving a LN slab of 400 nm thick. The electrode pattern was defined along the microcavity using electron beam lithography alignment, followed by electron beam evaporation and lift-off process. To facilitate efficient light coupling, the coupling waveguide was adiabatic tapered to a width of 3.5 $\mu m$ at the end facet over a length of 300 $\mu m$. Nufern UHNA-7 fiber whose mode field diameter is about 3.2 $\mu m$ at 1550 nm was used for coupling light into the chip. After the chip was diced for tests, a drop of UV adhesive was manually applied between the waveguide facet and a custom-built fiber holder, and the UV adhesive was cured for 5–10 minutes using a UV lamp. The package exhibited good long-term stability on coupling efficiency, which changed less than 3% over six months.

### Experimental setup
PDH locking was achieved by modulating light from a laser (ECDL: Santec TSL-550C, FL: Orbits Lightwave) with a local oscillator (Agilent N5181A) around 3 GHz. The drive frequency $\Omega_m$ was finely tuned to enhance the error signal (replacing the phase shifter commonly used in PDH detection). The output is detected by a high-speed PIN PD (EOT ET-3500AF), and mixed with the local oscillator to yield the PDH error signal. A bias-Tee separates the low frequency (DC) and high frequency (RF) signals. The former was fed to a servo for loose locking. And the RF signal is the sensing output, that is recorded by an oscilloscope and an electric spectrum analyzer.

## Data availability
The source data for all the figures are provided with the paper. All other data used in this study are available from the corresponding authors upon request. Source data are provided with this paper.

## Code availability

The code that supports findings of this study are available from the corresponding author upon request.

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

## Acknowledgements

We thank Prof. Qi-Fan Yang at Peking University for equipment loan and discussions. This work is supported by the National Natural Science Foundation of China (52022044 (C.Z), 62175127 (C.B), 62250071 (C.B)), the National Key R& D Program of China (2021YFB2801200 (C.B), 2022YFB3206801 (C.Z)), the Tsinghua University Initiative Scientific

Research Program (20221080069 (C.B)), and the Tsinghua-Toyota Joint Research Fund (C.B).

## Author contributions

X.M., C.Z, C.B., R.Z. conceived the project. X.M. and X.L. fabricated the devices; X.M. led the measurement and analysis with assistance from Z.C., Z.Z., K.L., B.C., J.H., C.Y.; X.M. and C.B. wrote the paper with inputs from all the authors. The work was supervised by C.Z., C.B. and R.Z.

## Competing interests

The authors declare no competing interests.

## Additional information

**Peer review information** : *Nature Communications* thanks the anonymous, reviewers for their contribution to the peer review of this work. A peer review file is available.

