## [Peer Review File · Nature Communications]

REVIEWER COMMENTS

Reviewer #1 (Remarks to the Author):

This manu. described an integrated E-field sensor based on TFLN. A high-Q micro-ring cavity with a dipole antenna and a laser frequency locking scheme, i.e., Pound-Drever-Hall (PDH) method, was employed to largely improve the sensitivity of the device to a record of $5.2 \mu\text{V}/\text{m}/\text{Hz}^{(1/2)}$. The micro-ring devices have been widely used for integrated sensors. The PDH method is also a well-established scheme to lock the wavelength of a laser to a reference cavity. Using this type of setup for improving sensitivity has also been proven in e.g., ref 26. But still, the demonstrated results in this manu. could be interesting for E-field sensor society. The sensing system in the manu. is rather complete, and can be integrated on a chip as the authors suggested. The main flaw of this manu. is that the overall concept is not very new, as well as the lack of some experimental details. I can recommend publishing it at NC, as long as the authors can cover the above issues. Some detailed comments are as follows:

1. The chip adopts a ring cavity. I think the detailed parameters of the cavity are missing in the manu., such as the diameter of the ring, the width of the waveguide, etc.
2. A dipole antenna is used to improve the sensitivity. How much gain in the E-field can be obtained with this antenna? What is the bandwidth of this antenna? Has the bandwidth of the antenna been considered in eq. (1) and Fig. 3d? How will it affect the bandwidth of the whole device?
3. In Fig. 3c, the dynamic range of the device is measured. There exists a saturation in the sensor response when the applied E-field is strong. What is the cause for this saturation? Can the dynamic range be improved further?
4. What are the interferogram signals shown in Fig. 2b and 2c? How are they measured?
5. Will the polarization of the E-field affect the sensor response? How is the E-field at 1-3m away from the emitting antenna calibrated?

Reviewer #2 (Remarks to the Author):

In this paper, the authors combined the PDH scheme with high-Q thin-film lithium niobate (TFLN) microcavities and demonstrated microcavity electric field sensors (MEFSs). The PDH scheme and the enhanced electro-optical interaction in the high-Q TFLN microcavity allow the integrated MEFSs to reach a sensitivity of $5.2 \mu\text{V}/(\text{mV}/\text{Hz})$, a sensing dynamic range of 123 dB, and a 3 dB bandwidth of 0.4 GHz, which is impressive.

1. The authors mentioned that “two of the devices were measured to have intrinsic Q-factors of 2.0 million and 0.4 million after packaging”. What is the reason for the reduction of the Q-factor.
2. The pictures of the fabricated chip should be shown in the manuscript.
3. Is there any special reason for choosing the LN thickness as 600 nm?
4. The authors use a ring resonator for the sensing. Note that LN is anisotropic. Would the sensitivity be improved further if using a racetrack resonator?
5. What is the chip-fiber coupling loss?
6. What kind of photodetectors used in the experiments? Would an APD be helpful for high sensitivity for the electric field sensing?
7. What is the limit of the sensitivity?
8. It would be great if the authors can give a discussion about the potential of the high-sensitivity sensors.

Response to manuscript NCOMMS-23-31172-T

We would like to thank the referees for the evaluation of our manuscript. The suggestions and comments from the referees have been addressed and are listed below. Revisions in the manuscript are indicated in brown texts.

Report of Reviewer 1 – NCOMMS-23-31172-T:

This manu. described an integrated E-field sensor based on TFLN. A high-Q micro-ring cavity with a dipole antenna and a laser frequency locking scheme, i.e., Pound-Drever-Hall (PDH) method, was employed to largely improve the sensitivity of the device to a record of $5.2 \mu\text{V}/\text{m}/\text{Hz}^{1/2}$. The micro-ring devices have been widely used for integrated sensors. The PDH method is also a well-established scheme to lock the wavelength of a laser to a reference cavity. Using this type of setup for improving sensitivity has also been proven in e.g., ref 26. But still, the demonstrated results in this manu. could be interesting for E-field sensor society. The sensing system in the manu. is rather complete, and can be integrated on a chip as the authors suggested. The main flaw of this manu. is that the overall concept is not very new, as well as the lack of some experimental details. I can recommend publishing it at NC, as long as the authors can cover the above issues. Some detailed comments are as follows:

Response) We thank the Reviewer for the careful report and recognizing our device have potential to reach high sensitivity in an integrated platform and can be interesting to the E-field sensing society. As for the novelty of this work, we have revisions as follows.

(1) Ref. 26 senses the strain signal within the PDH locking bandwidth, while our devices sense the signal well above the sensing bandwidth. The sensing signal is readout by beating the PDH locked laser with a frequency comb. In other words, the beating frequency was used as the sensing output. Although it reached an ultrahigh sensitivity, a relatively complicated system is needed. In our case, we sense the signal well beyond the PDH locking bandwidth. Thus, we rely upon the linear error signal slope for sensing and the voltage signal is used as the output directly. A relatively simple setup can be used. Furthermore, our experiment is the first demonstration to use PDH locking and integrated TFLN microcavity for E-field sensing, to our

knowledge.

We compare the sensing method with ref. 26 in the ‘sensing principle’ section, see the end of page 2.

(2) We believe our theoretical analysis of the PDH sensing in the frequency domain and the derived Eq. 1 are also new. The derived equation is general to applications using the PDH detection for the sensing of other physical quantities. And the analysis method may inspire further work on other microcavity-based sensing approaches.

We emphasized the possible impact of the derived sensing formula for other sensing applications at the end of the introduction section, see page 2. We also extended the theory to accommodate the case of strong applied electric field in the revision (see revised Supplementary Sec. 1). We added numerical simulations to validate our theory in new Fig. S2 as well.

Additionally, we provided more experimental details in the revision (please see our response in the following).

Report-1) The chip adopts a ring cavity. I think the detailed parameters of the cavity are missing in the manu., such as the diameter of the ring, the width of the waveguide, etc.

Response-1) We thank the reviewer for the comment. We listed the cross-section geometry of the microcavity in Fig. 1b. In the revision, we pointed this out on page 2 (1st paragraph of the ‘sensing principle’ section).

In addition to the parameters illustrated in Fig. 1b, we have added other parameters of the microcavities as follows:

The width of the coupling bus waveguides is $1.2\ \mu\text{m}$. For Device 1, the diameter and waveguide-microcavity gap of the ring are $400\ \mu\text{m}$ and $1000\ \text{nm}$, respectively; while these parameters are $200\ \mu\text{m}$ and $850\ \text{nm}$ for Device 2 (see page 2, 2nd paragraph of the ‘sensing principle’ section).

As shown in Eq. 1, the performance of the MEFS is largely impacted by the Q-factor. Nevertheless, the design difference in the two samples should not lead to Q-factor difference, as bending loss and metal absorption loss with these diameters are negligible. The Q-factor difference is mainly because these two devices were fabricated and packaged in different runs. We also pointed this out in the 2nd paragraph of the ‘sensing principle’ section.

Report-2) A dipole antenna is used to improve the sensitivity. How much gain in the E-field can be obtained with this antenna? What is the bandwidth of this antenna? Has the bandwidth of the

antenna been considered in eq. (1) and Fig. 3d? How will it affect the bandwidth of the whole device?

Response-2) We thank the reviewer for the questions. We didn't consider the bandwidth of the antenna in Eq. (1) and Fig. 3d. We carried out a simulation from 30 MHz to 10 GHz using the finite integration technique [1]. The triangle dipole antenna has a dimension of $5 \text{ mm} \times 2 \text{ mm} \times 1 \text{ }\mu\text{m}$ (length \times bottom width \times thickness), and is patterned on a silicon substrate. The field amplitude gain, defined as the enhancement of the electric field amplitude between the electrodes over the applied electric field amplitude in the simulation, is shown in Fig. R1.

Figure R1: **Simulated electric field amplitude gain of the antenna.** The antenna has a resonance frequency around 6 GHz. In the measured frequency range below 3 GHz, the antenna gain varies about 1.2 dB. The inset is a zoom in of the frequency range from 30 MHz to 3 GHz. The orange point indicates the gain for 100 MHz, which is used frequency in Fig. 3c of the main text.

The resonance frequency of the antenna is 6.6 GHz. The field amplitude gain at 100 MHz (tested frequency in Fig. 3c in the main text) is 24.1 dB. In the frequency range from 80 MHz to 3 GHz, the simulated field amplitude gain varies by 1.2 dB (see the inset of Fig. R1. The measured MEFS response in power (amplitude) rolls off by 30 dB (15 dB) for Device 1 or by 20 dB (10 dB) for Device 2 in this frequency range (see Fig. 3d of the main text). Therefore, the antenna frequency response can be neglected when considering the MEFS bandwidth.

In the revision, we gave the geometry of the antenna on page 2, added it

may provide about 24 dB gain for the sensitivity (beginning of the ‘sensing principle’ section, page 2), and pointed out that the antenna response does not impact the MEFS bandwidth strongly (page 6, end of the ‘sensing characterization’ section). We also added Supplementary Sec. 3 (including Fig. R1 as a new Fig. S4) to discuss the gain of the antenna.

Report-3) In Fig. 3c, the dynamic range of the device is measured. There exists a saturation in the sensor response when the applied E-field is strong. What is the cause for this saturation? Can the dynamic range be improved further?

Response-3) If the applied electric field is strong, the higher-order modulation sidebands, e.g., $\tilde{A}(\pm 2\Omega_s)$ becomes not negligible. Then, when solving $\tilde{A}(\Omega_s)$ in Eq. (S4), the term $\tilde{A}(2\Omega_s)$ should be considered. Thus, $\tilde{A}(\Omega_s)$ will have a nonlinear relationship with $\delta\omega$ (see new Eq. S19 in Supplementary Sec. 1). Furthermore, the depletion of the carrier $\tilde{A}(0)$ due to the emergence of the sidebands should also be considered (see new Eq. S21 in Supplementary Sec. 1). Consequently, there is a saturation for the MEFS and these higher-order sidebands and pump depletion limit the dynamic range.

The dynamic range can be improved by further suppressing the noise floor. In the paper, by using fiber laser rather than external cavity diode laser, the noise floor was decreased by 6 dB. Accordingly, the dynamic range should improve by 6 dB.

In the revision, we added that the higher-order sidebands and pump depletion is responsible for the observed signal saturation on page 5, right column. We further extended Supplementary Sec. 1 to include the carrier depletion and high-order sidebands in the theory. The new Fig. S2 in Supplementary Sec. 1 also further validates Eq. 1. And we added that the lower noise floor can help with the dynamic range on page 6 (first paragraph, left column).

Report-4) What are the interferogram signals shown in Fig. 2b and 2c? How are they measured?

Response-4) The interferogram signals shown in Figs. 2b, c are the output from the unbalanced fiber Mach-Zehnder interferometer (MZI) in Fig. 2a. It has a calibrated free spectral range of 50 MHz, and can be used to calibrate the laser frequency scan when measuring the microcavity linewidth and the Q-factor.

We pointed out the interferograms are measured from the output of the MZI, and are used to calibrate the laser scan, see the caption of Fig. 2.

Report-5) Will the polarization of the E-field affect the sensor response? How is the E-field at 1-3m away from the emitting antenna calibrated?

Response-5) Yes, the polarization of the E-field does affect the sensor response. This is primarily due to the polarization of the MEFS antenna and the anisotropy of the LN. In the paper, the transmitting antenna (thus, the exiting E-field) is set at horizontally polarized, which matches the polarization of the MEFS antenna.

The emitting antenna is placed 1-3 m away to generate different levels of electric field intensity. The E-field is calibrated using a receiving log-periodic antenna and a probe kit (AR Inc., FL7006) with various experiment configurations.

We added these points in the revision, see the end of page 4.

Report of Reviewer 2 – NCOMMS-23-31172-T:

In this paper, the authors combined the PDH scheme with high-Q thin-film lithium niobate (TFLN) microcavities and demonstrated microcavity electric field sensors (MEFSs). The PDH scheme and the enhanced electro-optical interaction in the high-Q TFLN microcavity allow the integrated MEFSs to reach a sensitivity of $5.2 \mu\text{V}/(\text{m}\sqrt{\text{Hz}})$, a sensing dynamic range of 123 dB, and a 3 dB bandwidth of 0.4 GHz, which is impressive.

Response) We thank the Reviewer for the careful report and positive evaluation of our work. We have address the comments in the following.

Report-1) The authors mentioned that “two of the devices were measured to have intrinsic Q-factors of 2.0 million and 0.4 million after packaging”. What is the reason for the reduction of the Q-factor.

Response-1) We thank the Reviewer for the question. The Q-factor difference is mainly because these two devices were fabricated and packaged in different runs. Although the diameter of Device 2 is of 200 μm , which is smaller than that of Device 1 with a diameter of 400 μm , the bending loss and metal absorption loss of the ring with the proposed parameters are negligible for these diameters. No noticeable change of the Q-factors was observed for all the 4 samples, before and after electrode fabrication.

We pointed that the two devices were fabricated and packaged in different batches on page 2 (1st paragraph of the right column).

Report-2) The pictures of the fabricated chip should be shown in the manuscript.

Response-2) We show a picture of the fabricated chip (taken by camera) and a microcavity captured by a microscope, see Fig. 1c in the resubmission. We also listed the cross-section geometry of the microcavity in Fig. 1b.

Report-3) Is there any special reason for choosing the LN thickness as 600 nm?

Response-3) The thickness of commercially available thin film LN (produced by NanoLN) is in the range of 300 nm to 900 nm. And the 600 nm thickness is one of the standard products. Considering the challenge of LN dry etching and the need to suppress high-order-mode propagation, the LN thickness of 600 nm could also be a well-balanced choice. Other thicknesses should also be used for MEFS, as long as they allow high Q-factor, near critical coupling and a large K/κ .

We added the wafer is from NanoLN and pointed out that other thickness can also work in the Methods section.

Report-4) The authors use a ring resonator for the sensing. Note that LN is anisotropic. Would the sensitivity be improved further if using a racetrack resonator?

Response-4) Yes, taking into account the anisotropic of LN, we designed a pair of half-rounded electrodes around the ring resonator on x-cut LN to achieve a high electro-optic modulation efficiency. If a racetrack resonator is used, the largest electro-optic coefficient r_{33} could be fully utilized along the straight waveguides. The sensitivity will be four times as high as it is now.

We have included it in the Discussions section (1st paragraph of the Discussions section, page 7), and ref. 37 also helps with the discussion.

Report-5) What is the chip-fiber coupling loss?

Response-5) The chip-fiber coupling loss per facet was about 5 dB and fiber-to-fiber insertion loss was about 10 dB after packaging. We added this loss at the end of the 1st paragraph of the ‘sensing principle’ section, page 2.

The coupling loss mainly comes from the mode mismatch between the fiber and LN waveguide and can be reduced by bilayer mode size converter (see the new ref. 38, added in the Discussions section on page 7).

Report-6) What kind of photodetectors used in the experiments? Would an APD be helpful for high sensitivity for the electric field sensing?

Response-6) We thank the reviewer for the suggestion. The photodetector PD3 (shown in Fig. 2a) used in the experiments belongs to a PIN photodetector, which model and vendor are ET-3500AF from EOT, Inc. We added that the used PD is a PIN detector in the Methods (2nd section, page 7).

When the PD input power is lower, the PD noise is dominant in the total noise floor (see Supplementary Section 2). The sensitivity of the MEFS can be improved by an APD, considering its higher responsivity and lower noise-equivalent-power (NEP). However, as the PD input power increases, the relative intensity noise (RIN) will dominate over the PD noise. In this case, using APD may no longer contribute to a higher sensitivity.

We added that using a PD with a lower NEP (e.g., APD) can help with improving the sensitivity, see the Discussion section, page 7.

Report-7) What is the limit of the sensitivity?

Response-7) The improvement of sensitivity of MEFS relies upon increasing K/κ in Eq. 1 and reducing the noise floor. The sensitivity of MEFS could be further improved by the following methods:

(1) Increase the coefficient K : By using a racetrack resonator on an x-cut LN to fully leverage the highest electro-optic coefficient of LN, the sensitivity of the MEFS will be four times as high as it is now. We included it in the Discussions section, page 7 (see also response to Report-4).

(2) Reducing loss κ and increasing the Q-factor: We included it in the Supplementary Fig. S2c. The reported intrinsic Q-factor for TFLN integrated microcavities reaches 10×10^6 (ref. 27), but the material limited Q-factor of thin film LN is about 1.6×10^8 [2]. This is the material limit for MEFSs on κ . We emphasized that further Q-factor improvement is possible, but will be ultimately limited by the material absorption (see page 7 and ref. 36, and Supplementary Sec. 2).

(3) Reduce the noise floor: The noise floor is currently limited the PD noise and laser RIN. Fundamentally, the noise floor should be limited by the shot noise of the detected light (see Discussions section, page 7).

Therefore, we believe that increasing the sensitivity by 50 times is feasible, that is a level of $100 \text{ nV}/(\text{m}\sqrt{\text{Hz}})$. This is better than the reported trapped ion result [3] but with a much simpler architecture (see the last sentence of

the 1st paragraph of the Discussions section, page 7).

Report-8) It would be great if the authors can give a discussion about the potential of the high-sensitivity sensors.

Response-8) High-sensitivity E-field sensors are of great use in both science and engineering. When integrated into an MEFS chip (shown in Fig. 5b), their potential applications become more extensive.

These scalable MEFSs can be deployed for laboratory and field measurements. They can be used to form a sensing array in upper atmosphere to detect fast radio bursts and may give insights into their origins [4], or to study laser-guided lightning behaviours [5] (see the 2nd paragraph of the Discussion section). They can also aid in the detection and locating lightning, which can contribute to thunderstorm forecasting [6, 7]. These sensors can also enable comprehensive studies of high-altitude transient electromagnetic events in the stratosphere and mesosphere to reveal their impacts on lightning origin and ionosphere [8]. They find practical use in electromagnetic compatibility testing for emission and immunity (see Fig. 1 and introduction). Additionally, they can serve as portable devices to ensure human health in specific work environments under electromagnetic exposure to avoid potential adverse effects including nerve stimulation and temperature elevation [9].

We added the points in the revised manuscript, please see the end of Discussions section on page 7, and new refs. 43-45 in the main text.

References

- [1] Marklein, R. *The finite integration technique as a general tool to compute acoustic, electromagnetic, elastodynamic, and coupled wave fields* (IEEE Press and John Wiley and Sons, New York, NY, USA, 2002).
- [2] Shams-Ansari, A. *et al.* Reduced material loss in thin-film lithium niobate waveguides. *APL Photonics* **7** (2022).
- [3] Gilmore, K. A. *et al.* Quantum-enhanced sensing of displacements and electric fields with two-dimensional trapped-ion crystals. *Science* **373**, 673–678 (2021).
- [4] Zhang, B. The physical mechanisms of fast radio bursts. *Nature* **587**, 45–53 (2020).

- [5] Houard, A. *et al.* Laser-guided lightning. *Nature Photonics* **17**, 231–235 (2023).
- [6] Haddad, M. A., Rakov, V. A. & Cummer, S. A. New measurements of lightning electric fields in florida: Waveform characteristics, interaction with the ionosphere, and peak current estimates. *Journal of Geophysical Research: Atmospheres* **117** (2012).
- [7] Leal, A. F., Rakov, V. A., Pissolato Filho, J., Rocha, B. R. P. & Tran, M. D. A low-cost system for measuring lightning electric field waveforms, its calibration and application to remote measurements of currents. *IEEE transactions on electromagnetic compatibility* **60**, 414–422 (2017).
- [8] Pasko, V. P. Recent advances in theory of transient luminous events. *Journal of Geophysical Research: Space Physics* **115** (2010).
- [9] International Commission on Non-Ionizing Radiation Protection. Guidelines for limiting exposure to electromagnetic fields (100 kHz to 300 GHz). *Health physics* **118**, 483–524 (2020).

REVIEWERS' COMMENTS

Reviewer #1 (Remarks to the Author):

The authors have reply to my comments and revised the manuscript accordingly. I do not have any more suggestions, and I think it can be published.

Reviewer #2 (Remarks to the Author):

The authors have addressed the reviewer's comments well and the manuscript can be accepted currently.